# Mental Health and Health-Related Quality of Life in German Adolescents after the Third Wave of the COVID-19 Pandemic

**DOI:** 10.3390/children9060780

**Published:** 2022-05-25

**Authors:** Justine Hussong, Eva Möhler, Anna Kühn, Markus Wenning, Thomas Gehrke, Holger Burckhart, Ulf Richter, Alexandra Nonnenmacher, Michael Zemlin, Thomas Lücke, Folke Brinkmann, Tobias Rothoeft, Thorsten Lehr

**Affiliations:** 1Department of Child and Adolescent Psychiatry, Saarland University Hospital, 66421 Homburg, Germany; eva.moehler@uks.eu; 2Department of Clinical Pharmacy, Saarland University, 66123 Saarbrücken, Germany; anna.kuehn@uni-saarland.de (A.K.); thorsten.lehr@mx.uni-saarland.de (T.L.); 3Medical Association, Westfalen-Lippe, 48151 Münster, Germany; markus.wenning@aekwl.de (M.W.); dr.thge.1@gmail.com (T.G.); thomas.luecke@rub.de (T.L.); 4Vaccination Center, 57072 Siegen, Germany; 5School of Education and Psychology, Siegen University, 57072 Siegen, Germany; rektor@uni-siegen.de (H.B.); ulf.richter@zv.uni-siegen.de (U.R.); alexandra.nonnenmacher@uni-siegen.de (A.N.); 6Department of General Pediatrics and Neonatology, Saarland University Hospital, 66421 Homburg, Germany; michael.zemlin@uks.eu; 7University Children’s Hospital, Ruhr University, 44791 Bochum, Germany; folke.brinkmann@rub.de (F.B.); tobias.rothoeft@rub.de (T.R.)

**Keywords:** COVID-19 pandemic, mental health, emotion, psychological symptoms, quality of life, adolescents, children, vaccination

## Abstract

Evaluations after the first and second waves of the COVID-19 pandemic in Germany showed an increase in mental health problems and a reduction in health-related quality of life (HRQoL). The aim of the study was to assess those aspects after the third wave of COVID-19 in adolescents who decided to receive a vaccination. In students aged 12–17 years recruited from schools in one German region, mental health (by the strengths and difficulties questionnaire, SDQ) and HRQoL (by KIDSCREEN-10) were assessed by both a self- and parental report. Data from 1412 adolescents (mean age 14.3 years, SD = 1.64) and 908 parents were collected. The mean self-reported HRQoL was T = 53.7 (SD = 11.2), significantly higher in boys than in girls and higher in younger (12–14 years) than in older (15–17 years) adolescents. In total, 18.7% of adolescents reported clinically relevant psychological symptoms, especially peer problems (23.5%), emotional problems (17.4%), and hyperactivity (17.1%). Comparing the present data to evaluations after the first and second waves of COVID-19, adolescents rated a higher HRQoL and reported less mental health problems after the third wave. After 1.5 years of living with the pandemic, adolescents have adapted to the changes in everyday life. Further, the relaxation of restrictions, better school organization, and the prospect of the vaccination may have increased optimism, wellbeing, and contentment, leading to declining but still alarming rates of psychological symptoms.

## 1. Introduction

Coronavirus disease 2019 (COVID-19) caused by the severe acute respiratory syndrome coronavirus-2 (SARS-CoV-2) is a highly infectious disease which was first detected in Wuhan, China in December 2019. The outbreak of COVID-19 was classified as a pandemic in March 2020 by the World Health Organization (WHO) [1]. The most common COVID-19 symptoms are fever, cough, fatigue, and shortness of breath, but gastrointestinal symptoms and myalgia have also been reported. The range of illness severity reaches from mild symptoms to a severe course with a mortality rate of about 1–2% overall [1]. Usually, children and adolescents have a much milder course of disease and a better outcome than adults; mortality is extremely low [2]. In late 2020, the first vaccinations were available against COVID-19. The German vaccination program started in December 2020 focused initially on vulnerable groups, such as the elderly and immune-deficient people. In August 2021, two vaccines against COVID-19 were approved for adolescents between 12 and 17 years in Germany [3].

In Germany, multiple waves of the pandemic were identified: wave one from March–May 2020, wave two from September 2020–February 2021, and wave three from March–June 2021 [4]. The pandemic led to nation-wide restrictions, such as school and kindergarten closures, contact restrictions, and economic shutdown. Working from home and home schooling were established, when possible, healthcare, and social support systems were reduced to a minimum and most leisure time activities (e.g., sport clubs, gyms) were cancelled. This caused tremendous changes to everyday life, especially in families and led to an increased rate of stress and mental health problems, as well as to an increase in domestic violence [5].

The first meta-analyses reported increased rates of psychological problems in children and adolescents after the outbreak of COVID-19, mainly internalizing symptoms as anxiety, depression, or post-traumatic stress [6,7]. In Germany, the nation-wide COPSY (‘Corona und Psyche’) study evaluated the mental health and health-related quality of life (HRQoL) of children and adolescents after the first and second waves of COVID-19.

The authors of the COPSY study compared the results to pre-pandemic data of the BELLA study, which assessed mental health and well-being in a German nation-wide representative sample of children and adolescents. They found a significant decrease in the HRQoL of about one standard deviation, as well as an increase in psychological problems for 19 to over 30% of children and adolescents affected [8,9].

The aim of our study was to assess mental health and HRQoL in adolescents after the third wave of COVID-19 in Germany. Adolescents who decided to receive a prioritized vaccination as part of a model project were included. The outcomes of the evaluation after the third wave will be compared to the COPSY evaluations after the first and second waves of COVID-19 in Germany, as well as to pre-pandemic data.

## 2. Materials and Methods

The present study is part of a model project that was conducted from July to September 2021 in the German region of Siegen-Wittgenstein offering a prioritized vaccination against COVID-19 for adolescents, young adults, and their caregivers. All students of 12 years and older from secondary level schools, vocational schools, and universities, as well as their accompanying caregiver were invited to receive a prioritized SARS-CoV-2 vaccination (BNT162b2 by Biontech/Pfizer). The start of the model project was announced in local newspapers and radios, and on the homepage of the Siegen University. All interested adolescents, young adults, and families could participate. Vaccinations were administered at the local vaccination center of the region Siegen-Wittgenstein and at the vaccination center of Siegen University.

All adolescents aged 12–17 years and their accompanying caregiver participating in the model project were invited to complete a survey including questionnaires on sociodemographic data, history of COVID-19 infection, and vaccination status of the adolescent and the family, as well as the adolescents’ motivational reasons for receiving a vaccination. Further, adolescents and caregivers were asked to complete two questionnaires on the adolescents’ mental health as well as their current health-related quality of life (HRQoL). Both questionnaires (SDQ and KIDSCREEN-10) were completed as a self-version by the adolescent and as a parental version by one parent. Participation in the survey was independent of receiving the vaccination. All participants and their caregivers were informed and provided written consent prior to their participation in the survey. The study was approved by the local ethics committee.

To assess psychological symptoms, the Strengths and Difficulties Questionnaire (SDQ) [10] was completed as a self-report by the adolescents, and as a proxy version by their parent. The SDQ is an internationally validated, standardized screening questionnaire for children and adolescents (age range 3–17 years) including 25 questions with each item having three answering options (‘not true’, ‘somewhat true’, and ‘certainly true’) coded with scores from 0 to 2. The scores are summed up for five scales (‘emotional symptoms’, ‘conduct problems’, ‘hyperactivity’, ‘peer problems’, and ‘prosocial behavior’; each scale including five items).

The total problem score (TBS) was calculated by summing up the first four scale scores, except the scale ‘prosocial behavior’ as this scale represents not problematic but positive behavior. The subscale scores and the TBS were divided into clinical (>90th percentile), borderline (80th–90th percentile), and average (<80th percentile) cut-offs according to German norms [11]. In addition, the scales ‘emotional symptoms’ and ‘peer problems’ were combined to the composite scale ‘internalizing problems’, and the scales ‘conduct problems’ and ‘hyperactivity’ to ‘externalizing problems’ [12].

The current HRQoL of the adolescents was assessed by the KIDSCREEN-10 completed as a self-report and as a parental report by the accompanying caregivers. The KIDSCREEN-10 is a reliable and valid instrument designed for populations aged 8 to 18 years to assess subjective health and well-being (HRQoL) efficiently [13]. The version comprises 10 items answered on a 5-point Likert scale summing up to a total global score. Reference norms are available for 11 European countries and are reported as T-values [14].

Statistical analyses were performed using IBM SPSS Statistics Version 25. First, we reported the descriptive data, i.e., non-parametric data were reported as relative and absolute frequencies, parametric data as means and standard deviation (SD). Next, we calculated the group differences for interval data with student’s *t*-tests or with Welch tests in cases of inhomogeneity of variance. The normality of sample distribution was assumed. Finally, comparisons to reference data (from wave one, wave two, or pre-pandemic) were calculated by one-sample *t*-tests and one sample Chi^2^-tests.

## 3. Results

The present results focus on the current psychological symptoms and quality of life described by the adolescents aged 12–17 years and their parents. A publication with a description of the sample including the history of COVID-19 infection and vaccination status as well as the motivational reasons is in preparation [15].

Data from 1477 adolescents were collected, of whom 1412 were in the age range of 12–17 years. The remaining 65 cases were excluded from the analyses, as age was not reported (*n* = 16) or was >18 years (*n* = 49). Of the 1412 cases, 771 were in the age range of 12–14 years and 641 were 15–17 years. The mean age was 14.3 years (SD = 1.64). In total, 695 of the adolescents were female (49.2%), 674 male (47.7%), and 13 diverse (0.9%). In 30 cases (2.1%), sex was not reported. Due to small sample sizes, the diverse group (*n* = 13) were excluded from the statistical analyses regarding sex group differences. Descriptive statistics (means, SDs, relative and absolute frequencies) of this group can be found in the Appendix A. In total, 16.8% (*n* = 237) reported a migrant background, i.e., the participant or at least one parent was born abroad.

In total, 940 caregivers answered the questionnaires. Of these, 32 were excluded (age of child not reported in *n* = 20; age >18 years in *n* = 12). Of the remaining 908 parents, the mean age was 46.0 years (SD = 5.98); in 22 cases, age was not reported. In 72.0% (*n* = 654) of cases, questionnaires were completed by mothers, in 22.4% (*n* = 203) by fathers, in 2.2% (*n* = 20) by other caregivers, and in 3.4% (*n* = 31) the person was not specified.

### 3.1. Self- and Parent-Reported HRQoL and Mental Health

The self-reported and parent-reported HRQoL and psychological symptoms of the total sample are outlined in Table 1. The HRQoL reported by adolescents and parents is similar, whereas the externalizing, internalizing, and total mental health scores of the SDQ were higher in self-reports than in parental reports. More adolescents reported clinically relevant symptoms in total (18.5 vs. 13.0%) and especially in hyperactivity (17.2 vs. 8.9%) than their parents. Parents reported more emotional (20.9 vs. 16.7%) and conduct problems (18.8 vs. 11.9%) assumed to be present in their children than those reported in the self-version of the SDQ.

### 3.2. Self- and Parent-Reported HRQoL and Mental Health in Different Age Groups

Table 2 shows the differences in HRQoL and psychological symptoms between the age groups 12–14 years and 15–17 years. The HRQoL was reported as significantly higher in the younger than in the older group, but only by the adolescents themselves, not by the parents. Regarding mental health, older adolescents reported significantly more internalizing symptoms, but less externalizing symptoms than the younger ones. The parents of younger youths report significantly more externalizing and total problems, but not internalizing symptoms.

### 3.3. Self- and Parent-Reported HRQoL and Mental Health in Boys and Girls

In Table 3, the differences in HRQoL and psychological symptoms between girls and boys are outlined. Girls report a significantly lower HRQoL, more internalizing and total symptoms, but also more prosocial behavior than boys. From the parents’ view, the responses show that boys present more externalizing but less internalizing symptoms than girls and have a significantly higher HRQoL.

### 3.4. Comparison to Outcomes after the First and Second Wave of COVID-19

The German population-based COPSY study assessed the HRQoL and mental health of children and adolescents during the first year of the COVID-19 pandemic by using the same instruments (KIDSCREEN-10 and SDQ, self- and proxy-version) as in the present study [8,9]. In the COPSY study, data from 1040 adolescents between 11 and 17 years (mean age = 14.33 years, 48.8% male) were evaluated, which is comparable to the present sample. After wave one (May-June 2020), the self-reported HRQoL in adolescents was T = 45.4 (parent-reported T = 42.4) [8]; after wave two (December 2020–January 2021) the self-reported HRQoL in adolescents was T = 44.8 [8]. In one-sample t-tests, the HRQoL T-value of the current sample is significantly higher than the HRQoL after wave one (53.7 vs. 45.4; *p* < 0.001) and after wave two (53.7 vs. 45.4; *p* < 0.001) in the self-reports, as well as in the parental report after wave one (51.8 vs. 42.4; *p* < 0.001).

Regarding mental health, 30.4% reported clinically relevant SDQ total scores after wave one and 30.9% after wave two [8,9]. Compared to the present data, one-sample Chi^2^-tests showed significant differences for both waves (wave one: 18.5% vs. 30.4%, *p* < 0.001; wave two: 18.5% vs. 30.9%, *p* < 0.001).

### 3.5. Comparison to Outcomes before the COVID-19 Pandemic

The German nation-wide longitudinal BELLA study also assessed mental health and HRQoL in children and adolescents, in multiple assessments from 2003–2006 (baseline) to 2014–2017 (11-year-follow up) using the SDQ and KIDSCREEN-10 [16,17]. The mean self-reported HRQoL was T = 53.4 and the mean parent-reported HRQoL was T = 53.6, while clinically relevant psychological problems were reported in 17.6% [8,16]. The current HRQoL did not differ significantly in self-reports (53.7 vs. 53.4, *p* = 0.291), but was significantly different in parent-reported values (51.8 vs. 53.6, *p* < 0.001) compared to pre-pandemic data. The frequency of mental health problems did not differ significantly between current and pre-pandemic reports (18.5% vs. 17.6%; *p* = 0.360).

## 4. Discussion

The present paper is one of the first reports to follow up on the COPSY study on quality of life and mental health in adolescents after approximately 1.5 years of living with the COVID-19 pandemic. All in all, the results suggest a change in psychological stress in youths during the course of the pandemic. Whereas COPSY and other authors show an increase in mental health problems and a decrease in the HRQoL after the onset of the pandemic [8,18], the present results show a similar level as in pre-pandemic times. The cross-sectional design prevents the identification of causal attributions, but it can be discussed whether stress adaptation processes, better pandemic management, and improved medical knowledge, as well as coping strategies and a significant reduction in lockdown measures during the summer have led to an increased capability of living with the virus and a growing ‘back to normality’ feeling in the population. In addition, the prospect of imminent vaccination may have had a positive influence on psychological well-being.

### 4.1. Health-Related Quality of Life (HRQoL)

The HRQoL reported by adolescents and parents is similar, both scores are in the range of an average score (T = 50, SD = 10), which means that the HRQoL is not limited in the present sample in general. However, age and sex differences in HRQoL occur, as it was assessed to be significantly higher in the younger than in the older group and higher in boys vs. girls. These results are in accordance with many former findings from population-based studies, which show a decrease in the HRQoL in older adolescents and especially in girls [19,20,21,22]. Explanations for these differences can be found in an earlier onset of puberty and hormonal changes in girls, which lead to a physical and social transition period, challenges with new behavior, and with coping with the environment and can therefore decrease perceived well-being. Further, the vulnerability for psychological symptoms is increased in female adolescents which may impact psychological well-being [7].

Regarding the changes of HRQoL after the onset of the COVID-19 pandemic, comparable data is rare. In a systematic review about the effects of COVID-19 on HRQoL in children and adolescents, only six studies from different countries worldwide were included [18]. Three studies reported a decrease in HRQoL, among them the German COPSY study [23]. Two did not find a significant change, and another did not report a change. The authors conclude that different aspects of COVID-19, e.g., lockdown and quarantine measures, social isolation/distancing, and school closures, have negatively impacted the quality of life in children and adolescents [18]. The difference to the present data is that the cited studies all were performed at the beginning of the pandemic, at the time when more strict arrangements were enforced in most countries. The current study was performed in summer 2021 in Germany, when infection numbers were low, restrictions were loosened, schools were open, and a vaccination was already available. The impact of COVID-19 on adolescent life was not as strong as the year before, which could have led to an increase in quality of life again. Therefore, our data indicate a partial reversibility of the negative first impacts of lockdown on children.

### 4.2. Mental Health Problems

The present results show a moderate frequency of clinically relevant psychological symptoms in total (18.5% by self-report) which is similar to the pre-pandemic comparison rates used in the COPSY study and is also consistent with the results from a meta-analysis finding a pooled prevalence of psychological disorders in 17.6% [9,24]. Other findings promote lower rates using stricter criteria [25]. One must consider that the present evaluation used only questionnaire data and a less strict cut-off in the SDQ (80th percentile) to enable comparison with the COPSY data, which could have overestimated the real prevalence. Further, our data only report symptoms, not rates of manifest psychiatric disorders.

Age and sex-specific differences are shown, as older adolescents reported significantly more internalizing symptoms but less externalizing symptoms than the younger ones, girls report more internalizing symptoms, and parents report more externalizing symptoms in boys than in girls. This is in line with known data, e.g., that externalizing symptoms, e.g., ADHD or hyperactivity are more common in younger children and decrease with age [26]. On the other hand, internalizing symptoms such as anxiety or depression have a higher incidence in females [27].

The rates of mental health problems in the present study are lower than those reported in other publications after the outbreak of COVID-19. In the German COPSY study, prevalence rates increased to over 30% after wave one and two [8,23]. Additionally, the first meta-analyses support the data, reporting increased rates especially of depression (25–29%), anxiety (20–26%), sleep problems (44%), and post-traumatic stress symptoms (48%) [6,7].

Many of the cited authors consider lockdown measures, social distancing, and schooling from home to account for the increased rates, as they impact the social life and development of young people. This is supported by a recent study that found a negative association between lockdown measures and the mental health of children [28]. Although the assessment tools differed between studies, the direction of the results is clear. Again, the difference to the current data lies in the date the studies were performed. All the cited studies assessed mental health in the first year of the COVID-19 outbreak in which nearly all countries introduced exceptional circumstances and special regulations. The present data were collected at a time with less active regulations and when a vaccination was available to prospectively lower the restrictions.

Moreover, the reduced rates of psychological problems may be a result of an adaptation process to stress factors that has taken place. In the transactional stress theory of Lazarus, stress is defined as a relationship between an individual and the environment which can be influenced by two mediators, ‘cognitive appraisal’ and ‘coping’ [29]. Whereas the appraisal of a stressful situation (e.g., the COVID-19 outbreak) is based on factors such as expectancies, individual dispositions, predictability, and controllability and influences the individual’s stress reaction, coping includes cognitive and behavioral strategies to manage, tolerate, or reduce the stress reaction [29].

The increased rates of mental health problems and reduced HRQoL in the first year of COVID-19 can be interpreted as a reaction to the stress induced by worrying about health and unpredictable changes in the social, economic, and private life of the population. This is consistent with the impact of phase one and two of an epidemic where restrictions in public and private life and closures of health services may increase psychological stress, but also lower the possibilities of receiving help, e.g., from healthcare systems [5]. Phase three is seen as the ‘return to normality’ phase, where re-organizing and re-establishing services and practices take place. In this phase, coping strategies and new rules (e.g., vaccinations, tests, wearing masks) have been developed and decrease perceived stress as they make the situation more predictable and controllable. This may have influenced the mental health and general quality of life of our sample. Further, the decision to get vaccinated can be interpreted as an individual coping strategy which in turn may increase self-efficacy and lower psychological stress.

### 4.3. Sample Characteristics

The present sample included adolescents aged 12–17 years from a specific German region who decided to receive a vaccination against COVID-19. Although vaccination willingness is relatively high in Germany overall [30], recent international studies show that the decision to receive a COVID-19 vaccination in children and adolescents depends on vaccine safety and efficacy, the perceived risk of infection transmission, and specific socio-demographic variables [31,32,33]. The willingness to receive a vaccination is increased in older adolescents, in those using more social media, and those having both parents vaccinated [32]; the willingness is decreased in those with higher distress over the effects [31], and in families with a lower income and a migrant background [33]. This is in line with the results of the present sample, as parental education level was high (>80% had a high or medium level of education) [15] suggesting a higher socio-economic status (SES) of the participating families. Further, 85–89% of the parents were already vaccinated [15] which increases the likelihood of their children getting vaccinated as well.

Regarding these results, the sample does not represent a population-based group, which may have influenced the outcomes on HRQoL and mental health. Adolescents from families with a higher SES tend to have a higher quality of life and a lower risk of mental health problems [34], which may have underestimated the frequency of psychological symptoms in the population. In addition, the decision to receive the vaccination may lead to more positive thinking and optimism in adolescents, as they expect more normalization of daily life and less restrictions.

### 4.4. Strengths and Limitations

This is one of the first studies assessing the HRQoL and mental health of adolescents in a large sample with over 1000 participants after the third wave of COVID-19 in a defined German region. The outcomes were measured via self- and proxy-report by valid and reliable, as well as internationally used, questionnaires that allow comparison with other publications. The sample was not population-based but representative regarding age and sex and comparable to the adolescent group tested in the COPSY study after waves one and two. One limitation is the likely higher SES of the participants in the study, as only those willing to receive a vaccination were included. Therefore, our results can be generalized to samples that are vaccinated or intend to be, but not to the general population.

Moreover, one has to consider that mental health problems were assessed by questionnaires and not by clinical experts, which is why only psychological symptoms can be reported, not psychiatric diagnoses. Due to the cross-sectional study design, causal attributions cannot be made.

## 5. Conclusions

This study aimed to assess mental health and HRQoL after the third wave of COVID-19 in adolescents who decided to receive a prioritized vaccination. The adolescents in the present study had lower rates of mental health problems and a higher HRQoL compared to research results assessed at the beginning of the pandemic. The outcomes are at a comparable level to pre-pandemic times. It can be concluded that the improvement of mental health and HRQoL derives from an adaptation process, as adolescents have learned to cope with the changes to everyday life which has lowered their anticipated stress. Furthermore, the decision to receive a vaccination can be seen as a personal coping mechanism which has increased optimism and raised hope. Finally, after 1.5 years of living with the pandemic, restrictions were loosened; schools are now better organized, allowing more normality and a presumably better quality of life than at the beginning of the COVID-19 outbreak, suggesting that the lockdown’s negative psychosocial impact on children and youths as reported before is in part reversible. However, the results of the present study also imply that in a pandemic or other exceptional situations which lead to dramatic changes in the everyday life of adolescents, mental health problems increase and can manifest for a longer time. In particular, school closures, the cancellation of leisure activities, and shutdowns are constraining measures and risk factors for many adolescents for developing long-lasting mental health problems. Preventive measures should include an increase in psychoeducation, as well as consultation and treatment possibilities for affected adolescents.

## Figures and Tables

**Table 1 children-09-00780-t001:** Self- and parent-reported HRQoL and psychological symptoms in the total sample.

	Self-Report (*n* = 1412)	Parental Report (*n* = 908)
HRQoL ^a^		
General HRQoL index, mean T values (SD)	53.7 (11.23)	51.8 (12.84)
SDQ ^b^, composite scale mean scores (SD)		
Externalizing	5.1 (3.18)	3.8 (3.34)
Internalizing	5.4 (3.58)	3.5 (3.39)
Total	10.5 (5.60)	7.3 (5.76)
SDQ, clinically relevant symptoms (>80th percentile) ^c^		
Emotional problems, % (*n*)	16.7 (245)	20.9 (159)
Conduct problems, % (*n*)	11.9 (166)	18.8 (143)
Hyperactivity, % (*n*)	17.2 (239)	8.9 (68)
Peer problems, % (*n*)	22.7 (315)	24.8 (189)
Prosocial behavior, % (*n*)	8.9 (125)	7.9 (60)
Total, % (*n*)	18.5 (258)	13.0 (99)

^a^ 10-item general HRQoL index assessed by the KIDSCREEN-10; ^b^ strengths and difficulties questionnaire; ^c^ % of adolescents having a score in the ‘borderline’ (>80th) or ‘abnormal’ (>90th percentile) range.

**Table 2 children-09-00780-t002:** Self- and parent-reported HRQoL and psychological symptoms in different age groups.

	Adolescents 12–14 Years	Adolescents 15–17 Years	Significance ^c^
Self-report	*n =* 771	*n* = 641	
HRQoL ^a^			
General HRQoL index, mean T values (SD)	55.3 (11.06)	51.8 (11.17)	<0.001 ***
SDQ ^b^, mean scores (SD)			
Prosocial behavior	8.2 (1.84)	8.3 (1.74)	0.293
Externalizing	5.5 (3.34)	4.8 (2.96)	0.002 **^d^
Internalizing	4.9 (3.54)	5.9 (3.55)	<0.001 ***
Total	10.2 (5.79)	10.7 (5.36)	0.107 ^d^
Parent-report	*n* = 547	*n =* 361	
HRQoL ^a^			
General HRQoL index, mean T values (SD)	51.4 (12.96)	52.4 (12.64)	0.344
SDQ ^b^, mean scores (SD)			
Prosocial behavior	8.4 (1.70)	8.4 (1.84)	0.661
Externalizing	4.1 (3.47)	3.4 (3.08)	0.004 **
Internalizing	3.6 (3.51)	3.4 (3.20)	0.484
Total	7.7 (5.90)	6.7 (5.47)	0.032 *

^a^ 10-item general HRQoL index assessed by the KIDSCREEN-10; ^b^ strengths and difficulties questionnaire; ^c^ student *t*-tests if not otherwise specified; ^d^ Welch test due to inhomogeneity of variance. Significant results are outlined with asterisks (* = *p* < 0.05, ** = *p* < 0.01, *** = *p* < 0.001).

**Table 3 children-09-00780-t003:** Self- and parent-reported HRQoL and psychological symptoms in boys and girls.

	Girls	Boys	Significance ^c^
Self-report	*n =* 695	*n =* 674	
HRQoL ^a^			
General HRQoL index, mean T values (SD)	51.5 (10.70)	56.2 (11.22)	<0.001 ***
SDQ ^b^, mean scores (SD)			
Prosocial behavior	8.4 (1.79)	8.1 (1.78)	<0.001 ***
Externalizing	5.2 (3.25)	5.0 (3.13)	0.338
Internalizing	6.3 (3.73)	4.4 (3.15)	<0.001 ***^d^
Total	11.5 (5.80)	9.4 (5.20)	<0.001 ***^d^
Parent-report	*n =* 414	*n =* 482	
HRQoL ^a^			
General HRQoL index, mean T values (SD)	50.6 (12.64)	52.9 (12.92)	0.023 *
SDQ ^b^, mean scores (SD)			
Prosocial behavior	8.4 (1.76)	8.4 (1.72)	0.803
Externalizing	3.3 (3.07)	4.3 (3.50)	<0.001 ***^d^
Internalizing	3.8 (3.40)	3.3 (3.37)	0.038 *
Total	7.0 (5.42)	7.5 (6.03)	0.207

^a^ 10-item general HRQoL index assessed by the KIDSCREEN-10; ^b^ strengths and difficulties questionnaire; ^c^ student *t*-tests if not otherwise specified; ^d^ Welch test due to inhomogeneity of variance. Significant results are outlined with asterisks (* = *p* < 0.05, *** = *p* < 0.001).

## Data Availability

The data that support the findings of this study are available from E. Moehler but restrictions apply to the availability of these data, which were used under license for the current study and so are not publicly available. Data are however available from the authors upon reasonable request and with permission of E. Moehler.

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
