# Peer review of "Mental Health and Health-Related Quality of Life in German Adolescents after the Third Wave of the COVID-19 Pandemic"

_children, 2022, doi:10.3390/children9060780_

Round 1
Reviewer 1 Report
General
Authors explore in their manuscript ‘Mental health and health-related quality of life in German adolescents
after the third wave of the COVID-19 pandemic’ the development of in MH and HRQoL in German adolescents by age and by gender. They also compare the results they found on MH and HRQoL with the pre-COVID-19 period. Such developments are understudied, however some minor work should be done before this manuscript can be published.
Title
-
Abstract
Background
-
Methods
Please explain me <mental health (by Strength and Difficulties Questionnaire, SDQ) and HRQoL (by KID- SCREEN-10) were assessed by self- and parental report> when exactly you used self-report and when parental report. Now the reader has to guess.
Results
Please add a sentence <Mean self-reported Mental Health …>
Conclusion
-
Key words
-
Introduction
Explain what means the <BELLA> study.
Methods
Sample
Shouldn't the following sentence be changed <All students 12 years and older> into ‘All students of 12 years and older’
Measures
About the SDQ: it is for anybody not living in Germany very much unclear how you calculated the SDQ: first you include the questions on ‘prosocial behavior’ (and come to 25 fields); later on you exclude them – but you don't say that in that case you use (as a result of that) 20 fields.
Statistical analyses
Please rewrite this section: First, we … . Then, we … . Next, we … . Finally, we … . The readership easier grabs what you did and in which order the Results will be shown.
Results
Mostly sentences like this <can be found elsewhere [15]> are written only in case when something can be found somewhere else; not when a manuscript is ‘in preparation’.
Discussion
-
Tables, Figures
-
References
Author Response
General
Authors explore in their manuscript ‘Mental health and health-related quality of life in German adolescents after the third wave of the COVID-19 pandemic’ the development of in MH and HRQoL in German adolescents by age and by gender. They also compare the results they found on MH and HRQoL with the pre-COVID-19 period. Such developments are understudied, however some minor work should be done before this manuscript can be published.
Response: Dear reviewer, thank you very much for giving us the opportunity to revise our manuscript. We have made changes according to your suggestions. Please find changes highlighted in the manuscript.
Methods
Please explain me <mental health (by Strength and Difficulties Questionnaire, SDQ) and HRQoL (by KID- SCREEN-10) were assessed by self- and parental report> when exactly you used self-report and when parental report. Now the reader has to guess.
Response: To clarify we included the following sentence to the methods section: “Both questionnaires (SDQ and KIDSCREEN-10) were completed as self-version by the adolescent and as parental version by one parent.” In the abstract, we made the following change: “In students aged 12-17 years recruited from schools in one German region, mental health (by Strength and Difficulties Questionnaire, SDQ) and HRQoL (by KIDSCREEN-10) were assessed by both, self- and parental report.”
Results
Please add a sentence <Mean self-reported Mental Health …>
Response: As the mean score of mental health can only be reported as a sum score (no standardized values), we prefer to report the % of clinically relevant psychological issues in the sample. We included the following into the abstract: “In total, 18.7% of adolescents reported clinically relevant psychological symptoms, especially peer problems (23.5%), emotional problems (17.4%) and hyperactivity (17.1%).”
Introduction
Explain what means the <BELLA> study.
Response: The following was changed in the introduction: “The authors of the COPSY study compared the results to pre-pandemic data of the BELLA study, which assessed mental health and well-being in a German nation-wide representative sample of children and adolescents.”
Methods
Sample
Shouldn't the following sentence be changed <All students 12 years and older> into ‘All students of 12 years and older’
Response: We changed the sentence as suggested.
Measures
About the SDQ: it is for anybody not living in Germany very much unclear how you calculated the SDQ: first you include the questions on ‘prosocial behavior’ (and come to 25 fields); later on you exclude them – but you don't say that in that case you use (as a result of that) 20 fields.
Response: Thank you for the comment, we tried to make it clearer. The subscale ‘prosocial behavior’ represents non-problematic and positive behavior and therefore is not included to the Total Problem Score. This was added to the methods section.
Statistical analyses
Please rewrite this section: First, we … . Then, we … . Next, we … . Finally, we … . The readership easier grabs what you did and in which order the Results will be shown.
Response: The changes were made accordingly.
Results
Mostly sentences like this <can be found elsewhere [15]> are written only in case when something can be found somewhere else; not when a manuscript is ‘in preparation’.
Response: You are right, as the other manuscript is not published yet, the sentence has to be changed. We included to the results section: “A publication with the description of the sample including history of COVID-19 infection and vaccination status as well as motivational reasons is in preparation [15].”
Reviewer 2 Report
This study deals with evaluations after the first and second wave of the COVID-19 pandemic in Germany that demonstrated a rise in mental health problems and a reduction of health-related quality of life (HRQoL). The authors aimed to identification of those factors after the third wave of COVID-19 in adolescents who decided to receive a vaccination. In this study mental health and HRQoL were assessed by self- and parental reports. Data from 1412 adolescents and 908 parents were collected. Comparison analysis showed that fewer mental health problems were reported in the third wave and following vaccination. The authors have proposed that a part of this effect can be related to adaptation after living in the pandemic for more than 1.5 years plus the positive effect of fewer restrictions, better school organization, and vaccination. The authors are encouraged to add these points in the revised version:
· It seems that the authors have checked the normality of data and based on that used parametric and non-parametric tests. Please add the test of normality in the statistic section.
· Did the authors perform any sub-analysis based on the gender of the participants?
· How can we use these data for plans, preventive strategies, or treatment? Please add at the end of the manuscript about this to give perspective and applicability to the findings from this study.
Author Response
Comments and Suggestions for Authors
This study deals with evaluations after the first and second wave of the COVID-19 pandemic in Germany that demonstrated a rise in mental health problems and a reduction of health-related quality of life (HRQoL). The authors aimed to identification of those factors after the third wave of COVID-19 in adolescents who decided to receive a vaccination. In this study mental health and HRQoL were assessed by self- and parental reports. Data from 1412 adolescents and 908 parents were collected. Comparison analysis showed that fewer mental health problems were reported in the third wave and following vaccination. The authors have proposed that a part of this effect can be related to adaptation after living in the pandemic for more than 1.5 years plus the positive effect of fewer restrictions, better school organization, and vaccination. The authors are encouraged to add these points in the revised version:
- It seems that the authors have checked the normality of data and based on that used parametric and non-parametric tests. Please add the test of normality in the statistic section.
Response: Dear reviewer, thank you for your comments and suggestions. Regarding normality of the data, statistic tests can be neglected as our sample sizes are large enough (n=1412 and n=908) for calculating t-tests based on the assumption of normality. In big samples (usually n>30) the sampling distribution tends to be normal (according to the central limit theorem, see e.g. Field, 2009). Therefore, tests of normality were not conducted, but data were checked visually (e.g. by q-q-plots). We included the following to the methods section: “The normality of sample distribution is assumed.”
- Did the authors perform any sub-analysis based on the gender of the participants?
Response: We conducted analyses between males and females regarding differences in HRQoL and various aspects of mental health. Table 3 shows the main analyses between gender groups in self- and parental report. See also section 3.3 for further details.
- How can we use these data for plans, preventive strategies, or treatment? Please add at the end of the manuscript about this to give perspective and applicability to the findings from this study.
Response: Yes, you are right, these implications should be outlined. We included the following: “The results of the present study implicate that in a pandemic or other exceptional situations, that lead to dramatic changes in everyday life of adolescents, mental health problems increase and can manifest for a longer time. Especially school closures, cancellation of leisure time activities and shutdowns are constraining measures and a risk factor for many adolescents for developing long-lasting mental health problems. Preventive measures should include an increase of psychoeducation, as well as consultation and treatment possibilities for affected adolescents.”